# Exploring Factors, and Indicators for Measuring Students' Sustainable Engagement in e-Learning

**Jeongju Lee, Hae-Deok Song * and Ah Jeong Hong** 

Department of Education, College of Education, Chung-Ang University, Seoul 06974, Korea;
sohotpop@cau.ac.kr (J.L.); ah454@cau.ac.kr (A.J.H.)
* Correspondence: hsong@cau.ac.kr

**Abstract:** The topic of engagement has been attracting increasing amounts of attention in the field of e-learning. Research shows that multifarious benefits occur when students are engaged in their own learning, including increased motivation and achievement. Previous studies have proposed many scales for measuring student engagement. However, very few have been developed to measure engagement in e-learning environments. Thus, developing an instrument for measuring student engagement in e-learning environments is the purpose of this study. The participants of this study were 737 Korean online university students. Initial items were designed based on the literature. The instrument items were reduced from an initial 48 to 24 items after obtaining expert opinion and then validity and reliability analysis. Exploratory and confirmatory factor analyses were also conducted. Six factors, including psychological motivation, peer collaboration, cognitive problem solving, interaction with instructors, community support, and learning management emerged in the 24-item scale. This scale is expected to help instructors and curriculum designers to find conditions to improve student engagement in e-learning environments, and ultimately prevent students from dropping out of online courses.

**Keywords:** e-learning; student engagement; measuring instrument; online education

---

## 1. Introduction

E-learning leads to positive learning outcomes, such as a high level of learning achievement and higher-order thinking abilities, because it allows for leaners to actively engage in learning anytime and anywhere [1,2]. Despite these advantages, one important problem in e-learning is the higher dropout rate [3]. Students tend to engage less in e-learning environments than in traditional learning environments because interactions between learners and teachers are reduced due to distance [4]. The distance between instructors and learners makes difficulty of rich communication that makes them participate persistently and efficiently in online learning, so they cannot continuously engage in online learning [5]. Online learning is also a very challenging environment for developing self-regulated capacities of learners, and learners who do not self-regulated in learning will face difficulties in engaging learning [6]. Thus, it has been reported that a primary reason for this high dropout rate is students' low engagement levels [7].

Student engagement is defined as the level of effort or interaction between the time or the learning resources that develop learning outcome and experience [8] When students are highly engaged in their learning, they can improve their academic achievement, such as critical thinking and grades, and then apply the acquired knowledge to real life [9]. Student engagement is also an indicator of the quality of education and whether active learning is taking place in classes [2]. Scholars agree that student engagement is fundamental to success in higher education [10]. They insist that students' active involvement and student engagement are essential in transforming higher education institutions into

sustainable enterprises. While a significant focus of campus sustainability requires student engagement, student engagement indicators for sustainability remain understudied. Given that student engagement is recognized as an important factor that positively affects learning and an indicator of the quality of education, an appropriate measuring instrument for student engagement is needed.

However, most instruments for measuring student engagement have been developed for face-to-face learning environments. For instance, the Engagement versus Disaffection with Learning (EvsD) scale measures student engagement and disaffection based on behavioral and emotional factors [11]. The National Survey of Student Engagement (NSSE) measures four factors: academic challenges, learning with peers, interaction with school institutions, and supportive learning environments [12]. The Student Engagement Instrument (SEI) measures emotional and cognitive factors, and the Motivated Strategies for Learning Questionnaire (MSLQ) measures engagement that is based on cognitive strategies [11]. The factors in these instruments include interaction, participation in class activities, attitude for learning, cognitive problem solving, and so on. However, these scales are limited, because they do not reflect the characteristics of engagement that are emphasized in the e-learning environment, which differ from those that are emphasized in the face-to-face learning environment.

Although several studies have examined engagement in e-learning environments, they are also limited in that the level of student engagement is mostly measured by behavioral indicators, such as the number of logins, the number of questions asked, lectures taken, articles that are posted on the bulletin board, and times that they participated in online discussions [13]. Others have modified instruments for measuring engagement developed for face-to-face environments, such as the NSSE, and they have adapted them to the e-learning environment [2,14]. However, such studies are also limited in that they apply many items that are specific to face-to-face environments to e-learning environments, and therefore are not able to reflect the unique characteristics of engagement in e-learning environments.

Thus, a measuring instrument for student engagement in e-learning needs to be developed to reflect the characteristics of the e-learning environment. For this purpose, the characteristics of the e-learning environment and student engagement factors in e-learning were examined. Subsequently, a suitable measuring instrument for student engagement in the e-learning environment was developed. Developing an instrument to accurately measure student engagement in e-learning environments is expected to help instructors in designing effective e-learning curricula and reducing the high dropout rate that is associated with e-learning courses.

## 2. Theoretical Background

### 2.1. Factors of Student Engagement

Engagement has been identified as an important antecedent of learning achievement [15–17]. Early studies defined student engagement as a single dimension of the behavioral aspect. Based on this perspective, engagement was simply defined as "students' participation in various activities related to learning" [18]. Mosher and MacGowan [19] emphasized the behavioral characteristics of engagement and defined it as "attitudes toward the learning program or participatory behavior" (p. 14). However, these definitions lack other dimensions, such as the recognition of learning and the learner's psychological state [20,21]. There are currently various definitions of student engagement. Hu and Kuh [22] defined engagement as "the amount of effort dedicated to educational activities that bring out ideal performance" (p. 555). Lewis et al. [9] defined engagement as "the extent to which learners' thoughts, feelings, and activities are actively involved in learning" (p. 251). Connell et al. [15] categorized student engagement into three categories: the behavioral type, such as persistent learning, effort, and sustained concentration in learning; the emotional type, such as interest in learning and excitement; and, the psychological type, such as preference for challenges, independence, and involvement in tasks. These variations in the definition of student engagement imply that student engagement extends from the behavioral aspect to the psychological and cognitive aspects, while

the scope of engagement is extended from learning activities in curriculum (e.g., learning time, effort, and strategy) to extracurricular learning activities (e.g., club activities, external activities, and volunteer activities).

As shown in the above definitions, student engagement consists of both behavioral and emotional dimensions. Marks [17] and Newmann [23] defined behavioral engagement as observable behavioral characteristics, such as the level of effort that is dedicated to learning or the level of learning achievement; they defined emotional engagement as learners' emotions about learning, such as interest, boredom, and happiness. Finn [24] explained that student engagement consists of behavioral factors (participation) and emotional factors (identification) in his presentation of the participation-identification model. The behavior factor represents an active attitude toward learning, such as asking questions or submitting assignments, and the emotional factor refers to the students' feelings toward learning, such as involvement in or a sense of belonging to the learning community. Furthermore, there are other types of engagement like cognitive, academic, and performance engagement. Cognitive engagement relates to learners' investment of thought, mental effort, or learning achievement strategies [8,20]. Psychological engagement is similar to emotional engagement [25]. Academic engagement can be explained by activities, such as time that is invested in learning tasks, task performance, grades, etc. Performance engagement is a related indicator to academic engagement. It reflects the level of learning performance, which is related to confidence in learning, grades, test scores, and so on [26].

### 2.2. Indicators that Characterize Student Engagement in the e-Learning Environment

What behaviors then can be expected of engaged learners? In a study on student engagement in face-to-face learning environments, engagement encompasses learners' cognitive, emotional, and behavioral reactions to educational activities [27]. Kahu [28] explain that that the types of engagement that can occur in the class are effort to learning, interest for learning, enthusiasm for the topic, sense of belonging to class, deep learning, self- regulation, and relationship with others. Burch, Heller, Burch, Freed, and Steed [29] suggested the types of engagement that can occur in the class include enthusiasm for the course, interest for learning, effort, invested energy for learning, concentration on learning, and attention to class. Hu and Kuh [30] suggested that the types of engagement that can occur in the class are task completion, learning efforts, communication with instructors, knowledge construction, application, and understanding. Heaven, Mark, Barry, and Ciarrochi [31] argued that the types of engagement that can occur in the classroom include task performance, attendance, interest in learning, and belongingness, and Abbott et al. [32] listed the additional indicators of learning satisfaction and passion.

Reviews of previous literature on student engagement suggest that the following behaviors are important indicators of student engagement in face-to-face learning environments [28–31]: learning effort, participation in class activities, interaction, cognitive task solving, learning satisfaction, sense of belonging, and learning passion. Learning effort factors are behaviors that learners learn themselves in their own time, such as doing homework, preparing for lessons before class, and studying after class. Participation in class activities refers to active intervention in class activities, which is related to attendance, presentation, asking questions, and expression. Interaction refers to communication between the professor and the learner about the learning contents, and it can be regarded as asking questions or asking for help with learning. Cognitive task solving refers to a learner's internal cognitive processes, such as knowledge formation, understanding, application, and memorization. Learning satisfaction is a psychological condition that includes interest in learning, expectations about learning, and enjoyment of learning. A sense of belonging refers to the degree of connection with friends and colleagues in the learning community. Finally, learning passion refers to possessing an active mindset when a learner learns, and it may manifest itself as mental energy in learning and a willingness to confront challenges in the learning process.

What are the learning behaviors of learners who are actively engaged in learning in e-learning environments? Indicators for measuring student engagement in e-leaning can be found in the learning behaviors of successful learners in e-learning environments. According to Golladay, Prybutok, and Huff [33], successful online learners discuss their learning with peers they and are motivated to learn, invest an appropriate amount of time to prepare for lessons, and can utilize the technology that is needed to take online classes. Dabbagh [34] insisted that online learners could establish their learning concepts themselves, use online learning technology easily, communicate with peers, learn in a self-directed manner, and have a sense of belonging to the learning community. Hong [35] studied the behaviors of excellent e-learners in Korea. The identified behaviors included planning a learning schedule, interacting with the instructor, learning collaboratively, constructing knowledge, applying their learning to real life, developing their own learning strategies, selecting learning contents, and having the motivation to learn. As suggested by Dixon [36], the factors of engagement in online learning included skills, emotion, participation, and performance. Skills are style of learning, such as studying regularly, listening and reading carefully, or taking a note. Emotion is state of feeling about learning, such as effort or a desire to learn. Participation is behavior in course, such as chat, discussion, or conversation. Performance is an outcome, such as grade or doing well on test.

These findings suggest several important characteristics of e-learning environments that can be considered to be indicators of student engagement. Successful and engaged online learners learn actively, have the psychological motivation to learn, use prior knowledge well, manage their learning schedule, and utilize online technology effectively. Moreover, they have great communication skills and are proficient in both cooperative learning and they are self-directed [34–36].

## 3. Methods

### 3.1. Context and Sample Characteristics

The populations in the study were college students at a four-year open university. This university was founded in 1972, which is the first lifelong educational institution in Korea and the first four-year national Open University in Korea [37]. It has one main campus that is located in Seoul and 13 local campuses. The total enrollment students are 113,780 and the number of professors is 152 in 2017. This institution has 22 majors in undergraduate schools and 18 majors in graduate schools. An online survey design was administered to collect the data. An email explaining the survey was sent to the school's staff members, asking them to invite their currently registered undergraduate students to voluntarily participate in the study. The students were asked to click on a link in the e-mail, which gave them access to the online questionnaire.

A total of 737 students were participated in this study. First, 218 students participated in the online survey for an exploratory factor analysis (EFA). Female participants comprised 61.5% of the sample, while males comprised 38.5%. The distribution by age was 8.7% for 20–29 years old, 34.7% for 30–39 years old, 38.1% for 40–49 years old, and 18.5% for over 50 years old. The distribution by year in university was 19.3% in the first year, 36.2% in the second year, 28.4% in the third year, and 18.1% in the fourth year. Subsequently, 519 students participated in this survey; a confirmatory factor analysis (CFA). Females comprised 68.8% of the sample, while males comprised 31.2%. The distribution by age was 10.8% for 20–29 years old, 28.6% for 30–39 years old, 38.7% for 40–49 years old, and 21.9% for over 50 years old. The distribution by year in university was 31.7% in the first year, 37.1% in the second year, 19.1% in the third year, and 12.1% in thee fourth year.

### 3.2. Scale Development Process

To develop a measuring instrument of student engagement in e-learning environments, we followed the scale development process that was proposed by DeVellis [38]. According to DeVellis' process, this study employed a five-stage process.

Setting the Item Pool. The related literature was reviewed to develop the pilot items [2,15,32,34,35]. We identified a definition of student engagement and its factors and then analyzed engagement behavior after examining research on distant education, online learning, etc. Six factors with 48 items were developed at the initial stage.

Ensuring Content Validity. To identify the validity of the factors that were developed in the preliminary research, the measurement instrument was reviewed by five experts in educational technology and e-learning. Factors that were not appropriate were modified during this step. In addition, the abovementioned experts in educational technology as well two experts in student engagement and educational psychology reviewed the validities of the items in each factor. During this step, the experts reviewed whether the items belonged to the factors and whether the items effectively measured student engagement. Furthermore, they checked meaning overlapping among items. After that, the content validity ratio (CVR) of each item was calculated and the items whose CVR values were 0.75 or less were deleted. A total of 14 items were deleted and 34 items were left. As a result, two items were deleted. The items included, "I understand deeply the knowledge that learned from online classes" and "I want to achieve higher scores in the classes that I am taking".

Before conducting the EFA, we measured the descriptive statistics for each item and reviewed the minimum value, maximum value, mean, standard deviation, skewness value, and kurtosis value. The mean and standard deviation of each item should be 1.5–4.5 and over 0.75 [39]. Items whose standard deviations were below 0.75. were deleted. In addition, the skewness and kurtosis values should be below 2 and 4, respectively [40], and all of the items satisfied that standard. As a result, 32 of the original 34 items remained. Thus, the online questionnaire included 32 items on a five-point Likert scale. The Cronbach's α calculated to examine the reliability of the instrument was 0.938. Finally, the item-total correlation was calculated and the value of all the items was above 0.03, which is an acceptable level.

Implementation Stage. Item Analysis and Construct Validity. An EFA was conducted to examine the factor structures of the items. Prior to the EFA, the KMO measure of sampling adequacy and Bartlett's test of sphericity can be used to determine whether the sample that was used in the analysis was appropriate for factor analysis. The KMO sample adequacy measure was 0.92, which was greater than the reference value of 0.8. In addition, the probability of significance of the Bartlett sphere formation verification was 0.00. This means that there was a correlation between the independent variables, and each item had a common factor. After the EFA, a CFA was conducted to verify the suitability of the factor structures. In this step, the model fits were examined to explain whether the items of each factor really explained the factor. Indexes were identified, such as the comparative fit index (CFI) measure, the Turker-Lewis index (TLI) measure, the root mean square error of approximation (RMSEA) measure, the adjusted goodness of fit index (AGFI), the relative fit index (RFI), and the Standardized RMR (SRMR) measure. Cronbach's α was also calculated to measure reliability.

Testing the Reliability. To test the reliability of the instrument for measuring student engagement, the total-item test score correlation and Cronbach's α were examined. Cronbach's α values that were above 0.70 were accepted as an adequate indicator of reliability. The total-item test score correlation explains the relationship between each item and the total score of all the test items.

## 4. Results

The EFA was conducted using principal component analysis and the Oblimin method for the rotation of factors. In the EFA, seven items were removed by applying the standard of 0.4 for each item factor loading. As a result, six factors with eigenvalues of 1 or more were obtained. They were found to explain 64.8% of the total variance as seven factors.

Table 1 shows the results of the EFA. The items for Factor 1 are related to psychological motivation. Items in Factor 2 are related to peer collaboration or collaborative learning. Items in the Factor 3 are related to cognitive problem solving, which represents a type of cognitive learning process. Items in Factor 4 are related to interactions with instructors. Items in Factor 5 are related to community

support. Finally, items in Factor 6 are related to learning management in the e-learning environment. The Cronbach's α of the six factors were 0.913, 0.854, 0.819, 0.773, 0.827, and 0.909, respectively.

**Table 1.** Factor Loading and Item Total Correlation Values.

| Factor Items | Psychological Motivation | Peer Collaboration | Cognitive Problem-Solving | Interactions with Instructors | Community Support | Learning Management |
|---|---|---|---|---|---|---|
| Enjoying learning | **0.774** | −0.095 | 0.079 | 0.061 | −0.097 | 0.041 |
| Stimulating interest | **0.750** | −0.044 | 0.144 | −0.011 | −0.106 | 0.034 |
| Usefulness of the course | **0.725** | −0.037 | −0.009 | −0.033 | −0.045 | 0.197 |
| Satisfied with the course | **0.721** | 0.056 | 0.162 | −0.064 | 0.029 | −0.033 |
| Learning expectations | **0.704** | −0.102 | 0.027 | 0.210 | −0.055 | 0.093 |
| Motivation | **0.702** | −0.066 | 0.118 | 0.067 | −0.217 | 0.069 |
| Requesting help | −0.042 | **0.893** | −0.057 | 0.029 | −0.018 | 0.011 |
| Collaborative problem solving | −0.038 | **0.843** | 0.059 | 0.075 | −0.053 | −0.024 |
| Responding to questions | 0.111 | **0.813** | 0.075 | −0.084 | 0.195 | 0.022 |
| Collaborative learning | −0.006 | **0.811** | −0.011 | 0.110 | −0.150 | −0.032 |
| Collaborative assignments | −0.087 | **0.663** | 0.069 | 0.175 | −0.220 | 0.077 |
| Deriving an idea | −0.023 | 0.006 | **0.849** | 0.039 | 0.038 | −0.043 |
| Applying knowledge | −0.047 | 0.087 | **0.788** | −0.098 | −0.041 | 0.060 |
| Analyzing knowledge | −0.012 | 0.007 | **0.780** | 0.038 | 0.175 | 0.092 |
| Judging value of information | 0.136 | −0.034 | **0.703** | −0.047 | 0.035 | 0.026 |
| Approach with new perspective | 0.135 | −0.052 | **0.703** | 0.107 | −0.172 | −0.076 |
| Communicating with the instructor | 0.049 | 0.036 | 0.048 | **0.871** | 0.124 | −0.061 |
| Asking questions | −0.005 | 0.049 | −0.037 | **0.836** | 0.113 | 0.064 |
| Belonging to community | 0.314 | 0.222 | −0.049 | −0.047 | **−0.649** | 0.118 |
| Connection with peers | 0.341 | 0.223 | −0.030 | −0.067 | **−0.636** | 0.101 |
| Interaction with peers | −0.055 | 0.455 | 0.128 | −0.005 | **−0.503** | 0.114 |
| Self-directed study | −0.127 | 0.093 | 0.024 | 0.022 | −0.050 | **0.764** |
| Managing own learning | 0.063 | −0.158 | 0.045 | −0.010 | −0.242 | **0.761** |
| Managing own learning schedule | 0.041 | −0.056 | 0.147 | 0.062 | −0.039 | **0.664** |
| Eigenvalues | 11.235 | 3.633 | 1.982 | 1.522 | 1.19 | 1.186 |
| Explained variance (%) | 35.108 | 11.354 | 6.195 | 4.756 | 3.722 | 3.706 |
| Total explained variance (%) | 35.108 | 46.462 | 52.657 | 57.413 | 61.135 | 64.841 |

For the CFA of the student engagement scale for e-learning, a model that was derived from the EFA was set up and tested. The SRMR < 0.10, RMSEA < 0.08, CFI > 0.90, TLI > 0.90, AGFI > 0.85, and RFI > 0.85 were used to evaluate the overall fitness of the model. As shown in Table 2, all the goodness-of-fit measures met their respective criterion. This indicates that the proposed model fits the data reasonably well.

**Table 2.** Goodness-of-Fit Measures.

| SRMR (<0.10) | RMSEA (<0.08) | | | CFI (>0.90) | TLI (>0.90) | AGFI | RFI |
|---|---|---|---|---|---|---|---|
| | LO 90 | M | HI 90 | | | | |
| 0.0891 | 0.061 | 0.066 | 0.071 | 0.919 | 0.910 | 0.865 | 0.874 |

The standardized coefficients of the six factors that measure student engagement and all items were significant at $p < 0.001$. The CR value showed that all of the items satisfied the criterion for convergent validity, with values that were greater than the 1.965 level.

The factor loadings of the six factors are shown in Figure 1. The factor loadings for psychological motivation varied between 0.66 and 0.86; for peer collaboration between 0.65 and 0.82; for cognitive problem solving between 0.60 and 0.76; for interactions with instructors between 0.74 and 0.83; for community support between 0.52 and 0.94; and, for learning management between 0.57 and 0.88.

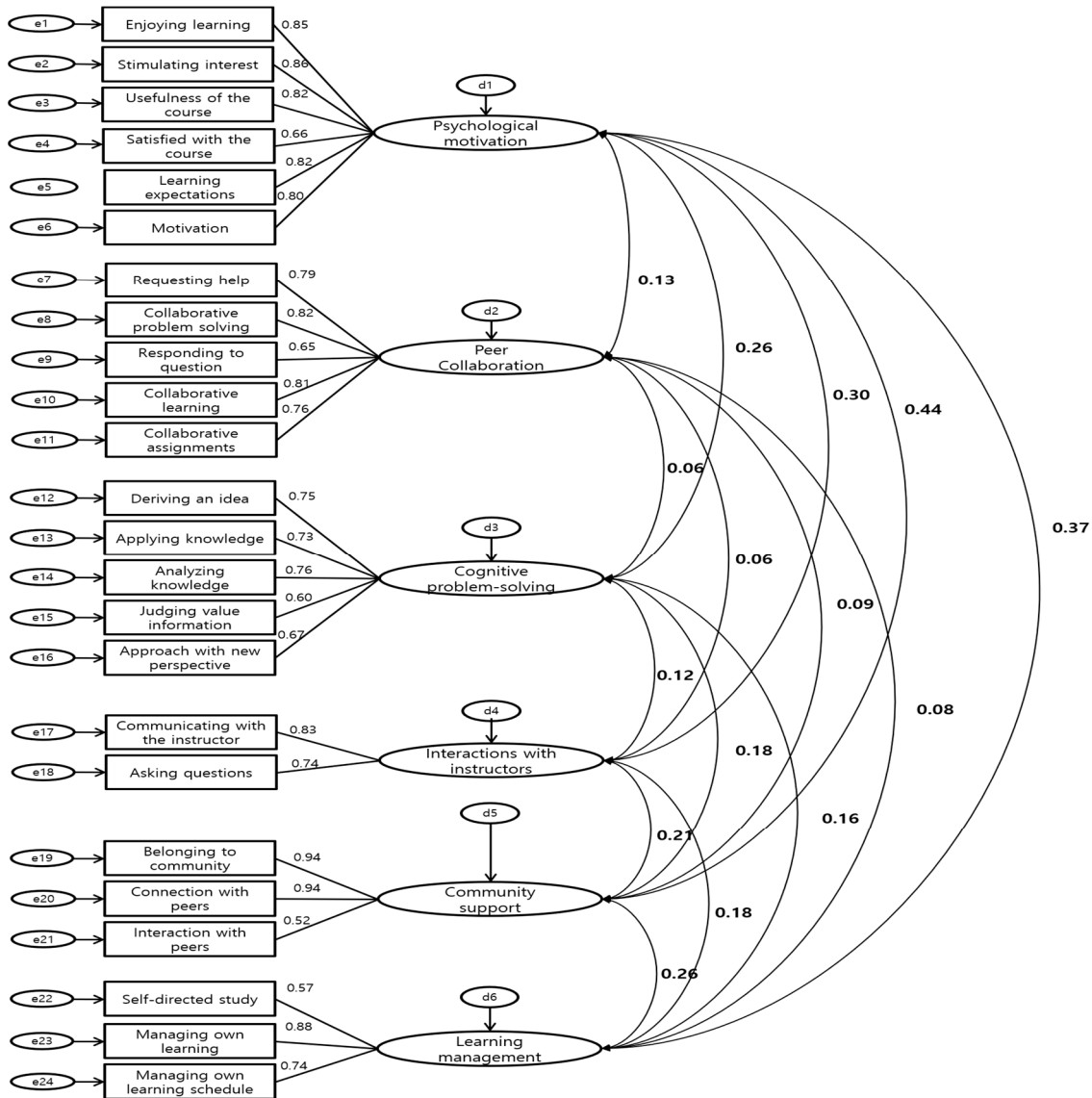

**Figure 1.** Confirmatory factor analysis results.

The final inventories consisted of six factors with 24 items (see Table 3). Psychological motivation included six items that are related to the psychological aspect of learning. Peer collaboration included five items that were related to collaborative learning activities with peers. Cognitive problem solving included five items that were related to internalizing cognitive tasks, two items that were related to interactions with instructors indicating learning-related communication activities between the learner and instructor, three items related to community support referring to psychological factors, such as perceived bonds and connections with other learners, and three items that were related to learning management indicating that the learner wants to actively participate in learning.

After removing the inappropriate items from the student engagement measurement for e-learning environments, a reliability analysis was performed on all 24 items. The Cronbach's α value of the developed factors was very high, at 0.93. The Cronbach's α coefficient was 0.896 for Factor 1 (psychological motivation), 0.876 for Factor 2 (peer collaboration), 0.825 for Factor 3 (cognitive problem solving), 0.758 for Factor 4 (interactions with instructors), 0.819 for Factor 5 (community support), and 0.717 for Factor 6 (learning management). Based on these results, it was clear that the reliability of the measuring instrument was acceptable.

**Table 3.** Factors and Items in Student Engagement in the e-Learning Environment.

| Factors | Items |
|---|---|
| Psychological motivation (6) | Online classes enhance my interest in learning.<br>I am motivated to study when I take an online class.<br>Online classes are very useful to me.<br>It is very interesting to take online classes.<br>After taking an online lesson, I look forward to the next one.<br>I am satisfied with the online class I am taking. |
| Peer collaboration (5) | I study the lesson contents with other students.<br>I try to solve difficult problems with other students when I encounter them.<br>I work with other students on online projects or assignments.<br>I ask other students for help when I can't understand a concept taught in my online class.<br>I try to answer the questions that other students ask. |
| Cognitive problem solving (5) | I can derive new interpretations and ideas from the knowledge I have learned in my online classes.<br>I can deeply analyze thoughts, experiences, and theories about the knowledge I have learned in my online classes.<br>I can judge the value of the information related to the knowledge learned in my online classes.<br>I tend to apply the knowledge I have learned in online classes to real problems or new situations.<br>I try to approach the subject of my online class with a new perspective. |
| Interactions with instructors (2) | I communicate with the instructor privately for extra help.<br>I often ask the instructor about the contents of the lesson. |
| Community support (3) | I feel a connection with the students who are in my online classes.<br>I feel a sense of belonging to the online class community.<br>I frequently interact with other students in my online classes. |
| Learning management (4) | I study related learning contents by myself after the online lesson.<br>I remove all distracting environmental factors when taking online classes.<br>I manage my own learning using the online system.<br>When I take an online course, I plan a learning schedule. |

## 5. Discussion

The purpose of the present study was to identify and develop indicators for measuring engagement in e-learning. The results showed that student engagement in e-learning was composed of six factors: psychological motivation, peer collaboration, cognitive problem solving, interactions with instructors, community support, and learning management. First, the psychological motivation factor represents learners' thoughts or feelings, such as interest, expectations, and motivation that is related to e-learning. Learning motivation and learning expectations are essential for higher level of learning activities in e-learning environments. This finding is consistent with previous studies that motivation and learning expectations are essential for problem solving activities in the e-learning environment [40,41]. It is also interesting that items regarding learning satisfaction belong to the motivation factor. While satisfaction refers to interest or satisfaction in the learning content in face-to-face learning environments, satisfaction in the e-learning environment reflects expectations and positive attitudes toward learning.

Second, the peer collaboration factor refers to activities in which learners discuss knowledge and collaboratively solve problems. Collaborative learning is a process of building and understanding knowledge with peers, and it is recognized as an important part of student engagement [31]. As collaborative learning and interaction is becoming increasingly important in the e-learning environment, it is significant that collaborative learning emerged as a separate factor in this study. This is further supported by the fact that the learning management system provided e-learners with more functions facilitating collaborative learning than those in the face-to-face learning environment.

Third, cognitive problem solving represents the process of acquiring, understanding, and utilizing knowledge. These are important factors because they affect learning achievement [41]. Items in this factor, such as approaching, structuring, analyzing, and applying knowledge, are consistent with cognitive process-related activities in three types of e-learning activities (e.g., absorb-type, do-type, and connect-type activities), as described by Horton [42]. Measurement in the face-to-face environment has mainly focused on behavioral or emotional types of engagement, and researchers have recently begun to pay attention to the cognitive process of learning [43]. Therefore, in this study, it is suggested that the cognitive aspects of learning, such as knowledge acquisition and processing, are emphasized as one factor, rather than the existing participation measurement tool.

Fourth, interactions with instructors shows the behavioral engagement in which the learner communicates with the instructor of an online course. In the e-learning environment, the level of engagement is higher when the learners sense a teaching presence that they feel in the actual learning field with the professor [44]. Teaching presence is facilitated when the learners communicate with instructors regularly [45]. Learners successfully learn when they feel a high level of teaching presence through continuous interaction with the instructor in e-learning courses [46]. Thus, interactions with the instructors seem to be the main factor in increasing learner engagement. Support behaviors and academic help also motivate learners and enhance their engagement in learning [47]. Therefore, the interactions with the instructors factor, which refers to communication acts, such as a requesting extra help from the instructor or asking questions regarding the contents of the lesson, can be considered as an important predictor of student engagement in e-learning.

Fifth, the community support factor is related to the psychological state of the learners, such as the bonds or the sense of community that is formed among learners that are enrolled in the same online courses. Emotional sense of belonging can be a major factor in the prevention of dropouts and help students to engage in classes. One reason for the high dropout rate is related to the lack of bonds or the sense of community among learners in online courses. If learners lack a feeling of connection or belonging with their fellow learners, then they tend to easily skip classes or leave them early, which may eventually lead them to drop out [24]. In other words, to increase the retention rate, instructors try to develop richer communication, such as net meetings to interaction, so that learners feel an emotional sense of belonging in the learning community [6]. Because of this reason, the importance of belonging has been emphasized by several studies [34,35]. However, the indicators for measuring engagement in e-learning with existing instruments are mostly quantitative measures of learning engagement, such as attendance and assignment submission, rather than learners' psychological or emotional engagement status. In light of this view, the community support factor that is derived from this study is significant, in that it is directly related to measures of the learner's psychological state.

Finally, learning management emphasizes behavioral engagement in which learners manage their own learning during active learning participation in online courses. This factor is related to active and self-directed learning activities for learners in an independent learning environment. According to Parkes, Reading, and Stein [48], engagement in the e-learning environment can appear as behavior characteristics, such as eliminating distractions in the environment during the online class, managing learning using the online system, and managing the learning schedule by taking a lecture plan when taking the online class. The indicators in this factor are different from behavior activities that are used in face-to-face learning environments. They include the number of logins, the number of lectures attended, the number of assignments submitted, presentation frequency, grades, and task performance, because they emphasize learner-initiated skills in managing online learning [16,31].

The findings of the present study are consistent with those of previous studies concerning the main factors of student engagement, given that student engagement is composed of behavioral, cognitive, and emotional engagement. Learning management and interactions with instructors are related to behavioral engagement, peer collaboration, and cognitive problem solving are related to cognitive engagement, and psychological motivation and community support are related to emotional participation. It is meaningful that we focused more on actual learning situations in the e-learning

environment and more intuitively subdivided the learner's specific engagement behavior, cognitive process, or learner's psychology extending from the existing three engagement factors.

## 6. Implications and Limitation of the Study

This study explored and identified the characteristics of student engagement in the e-learning environment. In comparison with the factors of student engagement measurement tools for face-to-face learning environments, interactions with instructors, cognitive problem solving, psychological motivation, and community support factors were common. However, the indicators for learning efforts are different. Learning effort, which in face-to-face environments measures the grade or task performance, and learning activity engagement, which measures attendance or participation, were not derived. On the other hand, in the e-learning environment, the learning management factor is related to activities, such as planning and management of learning and creating an effective learning atmosphere, and the peer collaboration factor, such as sharing knowledge and discussion, was derived. This means that the developed measurement tool in this study reflects the characteristics of e-learning, which emphasizes active and self-directed learning activities as an independent learning environment and emphasizes and supports collaborative learning.

There are several implications of these findings for online programs at both the curricular and class structure level. Online curricular should include courses that provoke engagement to educate students on effective engagement change strategies. In addition, within the classes, specific action for engagement should be integrated into online class lectures and assignments. For instance, engagement strategies could be incorporated into a review of online lectures, instructor feedback, and interaction between instructor and student.

Despite of potential strength of the proposed instrument, there are limitations in this study. One limitation is that this study used 737 students from one open education institution. Future studies should examine students from various open education institution to increase the validity of the instrument. The items that are derived from this study also require further validation. The reliability and validity of the developed items were verified through expert review, factor analysis, reliability, and convergent validity, but the validity of this study was verified based on the results of the data analysis. Future studies should further verify the measurement instrument's validity. Therefore, to verify the validity of the study results, follow-up research, such as verifying the predicted validity to test whether the developed test can predict engagement, is necessary.

**Author Contributions:** J.L. designed and wrote the draft. H.-D.S. modified and edited the paper and A.J.H. put forward the valuable suggestions.

**Funding:** This research was supported by the Ministry of Education of the Republic of Korea and the National Research Foundation of Korea (NRF-2017S1A3A2066878).

**Conflicts of Interest:** The authors declare no conflict of interest.

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
