# Peer review of "Exploring Factors, and Indicators for Measuring Students’ Sustainable Engagement in e-Learning"

_sustainability, doi:10.3390/su11040985_

Reviewer 1 Report

The purpose of this research is to develop an instrument for measuring student engagement in e-learning environments.  Although it is an  interesting and important article, the paper would be stronger if the following comments/suggestions are responded:

1. p. 1 “Exploring Factors and Indicators for Measuring 3 Students’ Sustainable Engagement in e-Learning” -- Sustainability is little-discussed in this article. Please consider explaining the followings: (1) sustainable engagement, (2) sustainable e-learning, and (3) sustainability of education.

2. p.5 ‘218 students from an open university located in Seoul participated in this survey. “  “519 students from an open university located in Seoul participated in this survey.”  However, the author(s) listed “736 Korean university students”  on p. 1 -- please reconfirm the number of participants.

3. The author(s) explained the data collection procedures on p. 5; however, please clarify the survey method of this research (e.g. web-based or face-to-face surveys).  In addition, please provide more details of the survey instrument.

4. p. 5 “As a result, two items were deleted.  “-- Meaning is not clear. Please consider providing more details about those two items.

5. p. 5  This research was conducted in "an open University located in Seoul.” -- Please consider adding “A case of …University” as the subtitle of this study.  In addition, please explain the study context of this open university (e.g. history of the distance programs, campuses, administration, admissions, departments, courses, faculty, and the number of students).

Please consider adding the following reference citation.

Choi, H., Lee, Y., Jung, I., & Latchem, C. (2013). The extent of and reasons for non-re-enrolment: A case of Korea National Open University. The International Review of Research in Open and Distributed Learning, 14(4). https://doi.org/10.19173/irrodl.v14i4.1314

6. Please consider providing more details about the management implications and the limitation of the study. This needs to be clarified.

7. Citations  & References -- Please check this section before submission. Please follow the journal guidelines.

Author Response

1. p. 1 “Exploring Factors and Indicators for Measuring 3 Students’ Sustainable Engagement in e-Learning” -- Sustainability is little-discussed in this article. Please consider explaining the followings: (1) sustainable engagement, (2) sustainable e-learning, and (3) sustainability of education.

 Response 1: Line 46-50. Added explanations on sustainability of education and its relation to engagement 

2. p.5 ‘218 students from an open university located in Seoul participated in this survey. “  “519 students from an open university located in Seoul participated in this survey.”  However, the author(s) listed “736 Korean university students”  on p. 1 -- please reconfirm the number of participants.

 Response 2: Line 18. 737 Korean online university.  

3. The author(s) explained the data collection procedures on p. 5; however, please clarify the survey method of this research (e.g. web-based or face-to-face surveys).  In addition, please provide more details of the survey instrument.

 Response 3: Line 175-178 Web- based survey was used. Survey design process was described. 

4. p. 5 “As a result, two items were deleted.  “-- Meaning is not clear. Please consider providing more details about those two items.

 Response 4: Line 206-207. Items were added. 

5. p. 5  This research was conducted in "an open University located in Seoul.” -- Please consider adding “A case of …University” as the subtitle of this study.  In addition, please explain the study context of this open university (e.g. history of the distance programs, campuses, administration, admissions, departments, courses, faculty, and the number of students).

Please consider adding the following reference citation. Choi, H., Lee, Y., Jung, I., & Latchem, C. (2013). The extent of and reasons for non-re-enrolment: A case of Korea National Open University. The International Review of Research in Open and Distributed Learning, 14(4). https://doi.org/10.19173/irrodl.v14i4.1314

Response 6: Although the samples were restricted to one open university, the purpose of this study was explore indicators that can be used for measuring online student engagement. Thus, the authors would like to keep the current title. The context of the open university was added Line 170-174. The recommended reference was added.

6. Please consider providing more details about the management implications and the limitation of the study. This needs to be clarified.

Response: Implications and limitations were added in Line 367-377.

7. Citations  & References -- Please check this section before submission. Please follow the journal guidelines.

Response 7: Citations and responses were revised according to journal guidelines. 

Reviewer 2 Report

It is good that you have sought to explore online engagement. A few areas need attention. You should check the spacing especially for the in text citations. For example, Line 35 space is missing between learning and (Leeds). Other formatting issues that need to be addressed are:

Lines 182 - 183: author initials in some of the references. 

Lines 198 - 199 male and female percentages add up to 100.05%

Line 278 decimal points missing in the values

FInally, it would be helpful to know the type of survey conducted - eg Likert scale. 

Author Response

Point1. You should check the spacing especially for the in text citations. For example, Line 35 space is missing between learning and (Leeds).

Response1: Revised spacing in text citations.

Point 2. Lines 182 - 183: author initials in some of the references. 

Response 2: Author initials were revised according to journal guideline.

Point 3. Lines 198 - 199 male and female percentages add up to 100.05%

Response 3: Lines 180-181  male and female percentage were revised.

Point 4. Line 278 decimal points missing in the values 

Response 4: Line 276 Decimal points were added.

Point 5. Finally, it would be helpful to know the type of survey conducted - eg Likert scale. 

Response 5:

Lines 175-178  A description on online survey process was added.

Line 213. A description on the types of survey was added: “Thus, the online questionnaire included 32items on a 5 point Likert scale.”